# Antimicrobial Activity and Mechanism of Functionalized Quantum Dots

**DOI:** 10.3390/polym11101670

**Published:** 2019-10-14

**Authors:** Keerthiga Rajendiran, Zizhen Zhao, De-Sheng Pei, Ailing Fu

**Affiliations:** 1College of Pharmaceutical Sciences, Southwest University, Chongqing 400716, China; keerthikesh@gmail.com (K.R.); w.joe@163.com (Z.Z.); 2Chongqing Institute of Green and Intelligent Technology, Chinese Academy of Sciences, Chongqing 400714, China

**Keywords:** quantum dots, antimicrobial efficiency, multi-drug resistance, reactive oxygen species (ROS)

## Abstract

An essential characteristic of quantum dots (QDs) is their antimicrobial activity. Compared with conventional antibiotics, QDs not only possess photoluminescence properties for imaging and photodynamic therapy but also have high structural stability. To enhance their antimicrobial efficiency, QDs usually are functionalized by polymers, including poly(ethylene glycol), polyethyleneimine, and poly-l-lysine. Also, QDs conjugated with polymers, such as poly(vinylpyrrolidone) and polyvinylidene fluoride, are prepared as antimicrobial membranes. The main antimicrobial mechanisms of QDs are associated with inducing free radicals, disrupting cell walls/membranes, and arresting gene expression. The different mechanisms from traditional antibiotics allow QDs to play antimicrobial roles in multi-drug-resistant bacteria and fungi. Since the toxicity of the QDs on animal cells is relatively low, they have broad application in antimicrobial research as an effective alternative of traditional antibiotics.

## 1. Introduction

The increasing prevalence of microbial infections and the rapid emergence of drug resistance to antibiotics created a critical health menace worldwide. An alarming rate of increase in mortality and morbidity in conventional therapeutics created an increased demand for an effective alternative agent for treating the infections [1]. Therefore, the finding of effective alternatives with antibacterial activity is now a research hotspot in biomedicine. 

Quantum dots (QDs) are a new type of fluorescent nanomaterial developed in recent years. Compared with traditional materials, QDs have unique physical and chemical properties, including high stability, an exceptionally narrow range of emission, and high quantum yield. Thus, they are widely used in biosensors, real-time tracking, multi-color labeling, and imaging [2,3]. Polymer-functionalized QDs imbue particles with more promising features and higher antibacterial activity [4,5]. Currently, functionalized QDs attract much attention because of their exceptional antibacterial mechanisms, indicating that QDs could be applied in antibacterial research as an effective alternative to traditional antibiotic drugs. In this review, we describe the mechanisms of functionalized QDs with regard to different types of microbes, as well as the biocompatibility of the QDs.

## 2. The General Antibacterial Mechanism of QDs 

The inhibitory action of QDs on microbial organisms mainly takes place via three molecular mechanisms (Figure 1): (1) destruction of cell walls/cell membranes; (2) production of reactive oxygen species (ROS) to destroy the cells; (3) binding with nucleic material (DNA/RNA) to inhibit cell proliferation. All the above processes can be evaluated using scanning electron microscopy (SEM), transmission electron microscopy (TEM), RT-PCR, and molecular dynamics studies. To increase their antibacterial efficiency, QDs are always functionalized using polymers and photosensitizers to induce more ROS and improve their attachment to bacterial components [6]. The effectiveness of the inhibition action differs in functionalized QDs based on their ligand, size, shape, zeta potential, and charge transfer effect (Table 1).

### 2.1. Destruction of Cell Wall/Cell Membrane

The QDs interact with the phospholipid bilayer, thereby roughening and shrinking the cell membrane. Furthermore, the cell is ruptured due to the direct attachment of the QDs, causing the discharge of cellular components [20]. The electrostatic interaction between the positive charge of QDs with the negative charge of cellular components creates membrane stress in the cell wall. The release of metal/ions into the cell increases toxicity inside the cell, resulting in cell death.

### 2.2. Production of ROS

Due to the high electron transfer, QDs produce a large number of free electrons and holes. The photoactivated QDs produce excessive free radicals, including hydroxide anions (OH^−^), superoxide singlet oxygen (O^2−^), triplet oxygen, and per-hydroxyl anions. The accumulation of ROS inside the cell inhibits respiration and replication, causing the cell death of microbes (Figure 1).

### 2.3. QD Interaction with Nucleus Components

Some QDs can enter and accumulate inside the nucleus. The interaction between QDs and nuclear content enables the inhibition of cell respiration, division, replication, and adenosine triphosphate (ATP) production. The nucleus damage ultimately results in cell apoptosis.

## 3. Antibacterial Activity of Graphene QDs

Graphene QDs (GQDs) are zero-dimensional materials converted to two-dimensional quasi-spherical materials of quantum size with a high edge effect and photostability [21]. GQDs have various structural dimensions, such as spherical, linear, pentagonal, and hexagonal. GQDs with a fine size and edge effect can be tuned and obtained through the attachment of different functional groups (ligands and polymers). Modified GQDs in structural orientations facilitate the attachment of GQDs to the microbial biomolecular lattice. Moreover, photostable GQDs, representing green and blue colors, contribute toward the bioimaging of the bacteria [22].

### 3.1. Antibacterial Mechanism of the Functionalized GQDs

Functionalized GQDs were shown to be effective antibacterial drugs [23]. The large π-conjugated system of GQDs easily attached to the cell wall of the bacteria via electron transfer. GQDs induced membrane stress by increasing oxidative stress; then, the ROS damaged the cell barriers, resulting in cell content leakage and cell death [24]. Nevertheless, non-modified GQDs have only weak antimicrobial effects. For example, Biswas et al. synthesized GQDs from micro-walled carbon nanotubes and studied their antimicrobial activity on *Escherichia coli, Bacillus subtilis, Pseudomonas aeruginosa,* and *Staphylococcus aureus* [25]. The results showed that the minimum inhibitory concentration (MIC) of the GQDs (~3 nm) was as high as 256 µg/mL for *E. coli* and *B. subtilis*, and approximately 512 µg/mL for *P. aeruginosa* and *S. aureus.*

Polymer-modified GQDs can increase the antibacterial activity since the polymers enhance the attachment of GQDs with the cell membranes of bacteria. For example, PEGylated GQDs exhibited 100% growth inhibition for *S. aureus* (25 μg/mL) and *P. aeruginosa* (50 μg/mL) following 8 h of incubation. Also, a synergistic effect could be produced in PEGylated silver GQDs, whose mechanism was associated with Ag^+^ binding with thiol groups (–SH) of the proteins and enzymes of cell barriers, resulting in cell disruption [8]. Moreover, polyvinylidene fluoride (PVDF)-modified graphene oxide QDs (GOQDs–PVDF) could be prepared as an antibacterial membrane. The GOQD-coated PVDF layer effectively inactivated *E. coli* and *S. aureus* cells, producing excellent antimicrobial activity (88.9% inhibition rate within 1 h) and anti-biofouling capability. The activity of the GOQDs–PVDF was higher than that of two-dimensional GO sheet- and one-dimensional carbon nanotube-modified membranes [26]. The significant antimicrobial and anti-biofouling properties were attributed to the strong interaction between the particles and thiol groups of the phospholipid layer of the cell membrane, subsequently inducing oxidation stress (Figure 2). Furthermore, GOQDs–PVDF possessed long-term stability and durability due to the strong covalent interaction between PVDF and GOQDs. This study suggested a new application of antimicrobial QD–polymer conjugates for potential wastewater treatment and biomolecule separation.

The photoluminescence property of GQDs in photodynamic therapy (PDT) adequately enables molecular tracking and labeling due to electron excitation from π–π orbitals and n–σ orbitals [27]. PDT can produce photoelectron excitation energy in the bactericidal process. The intrinsic and defective state emission of photoluminescence resulting from available valence electrons adds to the advantage of PDT. For example, amino-functionalized nitrogen (N)-doped GQDs were found to be an efficient photosensitizer, in which the five available electrons of nitrogen could create a high positive charge density on graphene carbons. Then, the N-doped GQDs could increase ROS production, leading to a more efficient bactericidal rate. As a result, N-doped GQDs led to 100% death of bacteria following 3 min of incubation [28].

### 3.2. Toxicity of GQDs on Mammalian Cells

GQDs are human-friendly particles utilized in various fields of biomedicine. There is no apparent toxicity of GQDs, regardless of the type of human cells treated with GQDs, whether in vitro or in vivo [29,30]. Biocompatible studies on GQDs identified that human cells showed more than a 90% steady live state of cell lines. Shang et al. suggested that no significant changes occurred in human neural stem cells after treatment with 250 μg/mL GQDs [31]. More than 80% of neuron cells, spinal cord cells, and cardiac progenitor cells survived upon treatment with 100 μg/mL GQDs. Furthermore, in vivo studies identified that graphene carbon QDs (40 mg) caused no apparent abnormal changes in mice. The body weight, kidney and liver function, and enzymatic biochemical reactions of mice were absolutely in the normal range after treatment with GQDs [32]. A hemolysis percentage of less than 1% was suggested, which is negligible in the medical context (˂5% is considered as a non-hemolytic percentage based on medical guidelines). Thus, GQDs can be used for infection treatment due to their high biocompatibility and biosafety.

## 4. Functionalized CdTe QDs against Multi-Drug-Resistant Bacteria and Fungi

CdTe QDs are hybrids of heavy metal ions in nano quantum confinement. CdTe QDs ranging in size from 2.5 nm to 8 nm exhibited antimicrobial activity against multi-drug-resistant bacteria [33]. The interaction of CdTe QDs is attributed to the same lattice dimension as that of nucleic acids and protein molecules in bacteria. Moreover, CdTe QDs possess electron transfer efficiency within bandgaps, producing photoluminescence, which is utilized in light-activated therapies; excitation of an electron from the valence band to the conducting band occurs along the 450–650-nm wavelength [34].

### 4.1. Mechanism of the QDs against Multi-Drug-Resistant Bacterial Strains 

CdTe QDs are widely used against multi-drug-resistant strains of bacteria and fungi. CdTe QDs release heavy metals when they enter the cytoplasm of microbial organisms, thereby retarding the growth of the organism. CdTe QDs inhibit microbial growth via three mechanisms: (1) toxicity of the heavy metal ion Cd^2+^ in the biological system (Cd^2+^ in the cytoplasm can accumulate in vesicle bodies, causing cytotoxicity to the bacterium); (2) production of oxidative stress (CdTe QDs downregulate the gene expression of superoxide dismutase (Mn-SOD) and endonuclease IV, responsible for the anti-oxidative property of the cellular system, as a result of the Cd^2+^ interaction with the genomes); (3) CdTe QDs attach to the lipophilic tail end of the phospholipid bilayer and damage the outer peptidoglycan layer of the cell wall. SEM and TEM were used to observe the smooth rod-shaped *B. subtilis* before treatment with CdTe QDs, showing that the bacterial cell walls were damaged after 4 h of incubation with the particles [35]. A concentration-dependent growth inhibition rate is widely observed in different species of microbes. Thus, the efficacy of CdTe QDs highlights their potential as an antibacterial candidate. 

Functionalized CdTe QDs have a higher antibacterial efficiency than non-coupled CdTe QDs. A synergistic effect of a CdTe QD–Rocephin complex against *E. coli* was exhibited because the –SHCH_3_COO– group of Rocephin and the –NH_2_ group of CdTe QDs interacted with the protons (H^+^) of organic biomolecules, in addition to the antibacterial activity of the QDs [10]. The strong electrostatic attraction exerted by the interaction of protons with the anionic –SHCH_3_COO– and –NH_2_ created an electrostatic force between the molecules, which induced a higher oxidative ability in the biomolecules and released several singlet and triplet oxygen radicals. About 0.5 μg/mL Rocephin along with 120 μg/mL CdTe QDs exhibited more than 90% growth of *E. coli* compared to 120 μg/mL plain CdTe QDs without Rocephin. Thus, the synergistic effect increased the growth inhibition property of CdTe quantum dots.

Moreover, rutin-conjugated TGA–CdTe QDs could significantly inhibit *E. coli* growth when the hydroxyl group (–OH) of rutin interacted with the CdTe QDs [36]. The rutin-conjugated TGA–CdTe QDs released OH^−^ and produced a positive hole in the valence band, which led to easy attachment of the rutin TGA–CdTe QDs through electron-hole transfer between the cell membrane and QDs. The OH^−^ produced both intercellular and intracellular oxidative stress in the *E. coli* through OH^●^ radical transportation via porins/pits between the membrane, thereby increasing the level of ROS. 

Polymer-coated CdTe QDs can increase protein-specific recognition [37]. Poly-l-lysine (PLL) is a biocompatible polymer that is also used to functionalize CdTe QDs to improve their bactericidal capacity. The inhibitory growth rate of PLL-coated CdTe QDs on *E. coli* depends on the number of PLL bilayers coating the CdTe QDs [38]. The intensity of luminescence emission increased with an increase in the number of PLL bilayers coating the CdTe QDs. The easy interaction with the PLL-coated QDs contributed toward the damage of the bacterial cell wall. The growth rate inhibition varied with the increase in the number of PLL bilayers. The growth rate of *E. coli* was completely inhibited using 8–10 PLL bilayers on QDs. The mechanism of inhibition was attributed to the disruption of the cell wall via interaction with the PLL/CdTe QD bilayer.

Photosensitizer-modified CdTe QDs carrying a photogenerated electron charge can produce ROS in PDT, which plays an important role in antibacterial activity on multi-drug-resistant bacteria [39]. The photosensitizer’s functional moieties control the quantitative effects of the potential redox value of the CdTe QDs. In *S. aureus, E. coli, Klebsiella pneumoniae,* and *S. Typhimurium*, light-activated photocatalytic experiments revealed that the CdTe QDs with 2.4 eV band and edge effects induced superoxide radicals through photoexcited electrons. The oxidative stress created by superoxide radicals through the light-activated mode was more effective, with 92% more death of bacteria than bacteria treated with control CdTe QDs.

The surface charge can affect the antibacterial activity of CdTe QDs [40]. The modification of 3-mercaptopropionic acid (negatively charged) on particles inhibited a larger number of *E. coli* than that CdTe QDs coated with the cysteamine (positively charged), evaluated using biological microcalorimetry. Positive QDs were prone to attaching on the surface of *E. coli*, subsequently destroying membrane structure and function. Also, positive QDs significantly increased membrane fluidity between bacteria and a dipalmitoylphosphatidylcholine model membrane, which enhanced the membrane permeability, resulting in intracellular contents leaking out of the cells.

### 4.2. Antifungal Mechanism of CdTe QDs

Han et al. revealed the inhibition mechanism of CdTe QDs on *S. cerevisiae*, which was attributed to intracellular accumulation of Cd^2+^, followed by the induction of cell dysfunction and deformation [41]. The cell wall shrank and enabled the creation of a passage for Cd^2+^ internalization following cell-wall corrosion. Moreover, the orange light emitted by CdTe QDs through photoelectron activation inhibited the growth rate of yeast (17.07 nm/L), whereas the yeast cell growth rate was inhibited with 18.01 nm/L using the green light emitted. Thus, the photosensitizing wavelength and the fluorescent color of the light radiated are also considered important physiochemical properties that decide the inhibitory rate of yeast with CdTe QDs. 

### 4.3. Toxicity of CdTe QDs on Human Cells

CdTe QDs, in general, are found to be toxic to human cells. The toxicity level depends on the concentration of the CdTe QDs in the drug system. However, functionalized CdTe QDs that are conjugated with biomolecules are human-friendly materials. For example, CdTe QDs coated with PLL had high biocompatibility on human cells following 12 h of incubation, showing no apparent toxicity on the cells [10].

## 5. Antibacterial Activity of CdSe QDs

Similar to CdTe QDs, other cadmium-based QDs maintain the same pattern of growth inhibitory activity on bacterial. The π–π conjugated electron transfer activates the Cd^2+^, thereby producing an enriched amount of ROS species inside bacterial species, resulting in growth inhibition. Priester et al. reported that CdSe QDs effectively impaired the growth of *P. aeruginosa*. Cd^2+^ ions entered the cell cytoplasm through pits and pores in the damaged cell wall, thereby facilitating the internalization and intracellular accumulation of Cd^2+^ ions. The increased quantity of heavy metal ions (Cd^2+^) increased the toxicity to the cell. The resulting toxicity restricted the cell respiration, multiplication, and reproduction of *P. aeruginosa*. Also, the accumulated ROS of oxygen molecules accelerated the increase in oxidative stress, resulting in rupture, deformation, and death of the cell. The MIC of inhibition of CdSe QDs was found to be 5 μg/mL for *P. aeruginosa*.

Hybrid particles of CdSe QDs/TiO_2_/graphene nanosheets showed high antibacterial activity due to their photocatalytic properties produced upon irradiation in the visible range against *E. coli* [14]. Face-to-face two-dimensional (2D) orientation of the nanographene sheets capped with CdSe QDs, featuring large π conjugation electrons, increased the movement of the photogenerated photons toward the bandgap and enhanced the photocatalytic property through inducing the high excitation energy of electrons. The growth retardation of *E. coli* was visibly attributed to photo radiation. The visible irradiation with CdSe QDs efficiently killed up to 90% of *E. coli*. The light irradiation-producing holes, as well as the increase in O^2−^ and OH radicals, increased the level of ROS, causing cell death.

Mei et al. reported the antifungal activity of CdSe QDs on yeast *S. cerevisiae* [42]. The direct attachment of Cd^2+^ onto the cell walls of *S. cerevisiae* disturbed cell integration. The high toxicity of the Cd^2+^ caused oxidative stress that increased the apoptosis-associated gene expression. A microcalorimetry study revealed that the growth rate decreased with the increase in concentration of Cd^2+^ inside the cells, whereas the inhibition rate increased with the increase in concentration of Cd^2+^ inside the cells. The CdSe QDs also reduced the mitochondrial potential of *S. cerevisiae*, resulting in the dysfunction of cellular metabolism.

## 6. Antimicrobial Activity of ZnO QDs

ZnO QDs range from 2 nm to 16 nm in size with a high superficial charge. Various studies on ZnO QDs ensured their biosafety and biocompatibility with regard to human cells [43,44]. The properties of ZnO QDs lead to their wide application in the treatment of bacterial and fungal infections.

### 6.1. Antibacterial Mechanism of ZnO QDs

Three plausible mechanisms are involved in the growth inhibition of ZnO QDs on microbial organisms: (1) electron–hole pairs (e^−^, h^+^) created by ZnO QDs induce ROS species of hydroxyl radical (OH·), superoxide anion (O^2−^), and per-hydroxyl radical (HO^2−^); (2) ZnO QDs directly rupture the phospholipid layer, leading to cell content and cytoplasm leakage; (3) Zn^2+^ accumulates in the cell, damaging the DNA and causing cell apoptosis [45].

The multi-drug resistance exhibited by bacteria against conventional antibiotics and therapy raises the need for a biocompatible alternative drug. Functionalized ZnO QDs have inhibitory ability against multi-drug-resistant bacteria. The antibacterial efficiency of ZnO QDs is dependent on their small size and surface modifications. For example, ZnO QDs with an anionic surface of acetate (CH_3_COO^−^) and nitrate (NO^3−^) groups was used for treatment against multi-drug-resistant *E. coli*. The MIC was found to be around 6 μL/mL in the presence of nitrate ZnO QDs (4–7 nm size) and reached 2.5 μL/mL after *E. coli* was treated with acetate ZnO QDs (3–5 nm size). The greater inhibition rate of acetate ZnO QDs was attributed to the greater anionic electrostatic force of the surface, which increased ROS production, leading to cell death [18].

When polymers functionalize the surface of quantum dots, the antimicrobial spectrum may be broadened. PEG-capped ZnO NPs inhibited both Gram-positive (*S. aureus*) and Gram-negative (*E. coli*) bacteria with an average size of 10 nm [46]. The antimicrobial activity may be attributed to the production of ROS and H_2_O_2_, which obviously inhibit microbial growth. Moreover, polyvinylpyrrolidone (PVP) was used to modify ZnO QDs to prepare a ZnO–PVP complex [18]. The complex produced strong effects against *Listeria monocytogenes, Salmonella enteritidis,* and *E. coli* O157: H7, resulting in a 5.3 log reduction of *L. monocytogenes* cells and a 6.0 log reduction of *E. coli* cells after 48 h of incubation, as compared to the non-capped ZnO QD control.

### 6.2. Antifungal Mechanism of the ZnO QDs

Fakhrouien et al. examined the antifungal activity of ZnO QDs against *Microsporum. gypseum* (*M. gypseum*), *Microsporum. canis* (*M. canis*), *Trichophyton. Mentagrophytes* (*T. mentagrophytes*), *Candida. albicans* (*C. albicans*), and *Candida. tropicalis* (*C. tropicalis*) [47]. The ZnO QDs entered the cytoplasm without causing significant damage to the fungal cell wall via phagocytosis. Then, the ZnO QDs accumulated in the vesicle and myelin bodies. Some ZnO QDs could enter the nucleus and attach to DNA, RNA, and chromatin, thereby damaging the nucleic material and causing cell death. ZnO QDs exhibited potent growth inhibition in all five species, compared with the clotrimazole control. The MICs of *M. gypseum*, *M. canis*, *T. mentagrophytes*, *C. albicans*, and *C. tropicalis* were 3.12, 1.56, 0.75, 12.5, and 12.5 μg/mL, respectively. All five inhibitory concentrations of ZnO QDs were much lower in comparison with that of the clotrimazole control (300 μg/mL).

## 7. Others

Silver indium sulfide (Ag/In/S–ZnS) QDs are hybrid nanomaterials with a shell–core structure, in which surface modification of ZnS by Ag/In/S QDs can produce extended fluorescence (˃300 ns). The particles retarded the growth of 90% *C. albicans* at a concentration of 125 μg/mL through oxidative stress [18]. Also, *N*-acetyl-l-cysteine-capped Ag/In/S QDs showed remarkable bactericidal activity (Gram-negative bacteria) at a concentration of 15 μg/mL [48], much lower than non-capped Ag/In/S QDs. Moreover, a recent study showed a novel nanocomposite derived from ZnO/CdS QDs embedded with cross-linked chitosan (cl-Ch-pMAc@ZnO/CdSQDs) [49]. The nanocomposites were fabricated under microwave irradiation through the conjugation of anionically functionalized chitosan cross-linked with ZnO/CdS QDs in the presence of diethylene glycol dimethacrylate as the cross-linker. The nanocomposite exhibited excellent antibacterial activity toward both *E. coli* and *B. subtilis*.

Composite materials used for antimicrobial purposes were found using transferrin-modified silver QDs coupled with zinc and rifampicin (Zn/RIF/Tf QDs). The Zn/RIF/Tf QDs were tested for their antibacterial activity against *Mycobacterium smegmatis* and *M. bovis BCG*. Zn/RIF/Tf QDs exhibited 10-fold higher antibacterial activity compared to the zinc and rifampicin complexes (Zn-RIF). The treatment of *M. smegmatis* and *M. bovis BCG*, with 1 μg/mL Zn/RIF/Tf QDs killed almost 99% of bacteria (Figure 3)**.** The higher efficacy of Zn/RIF/Tf QDs was attributed to the release of quantum-sized silver ions into the bacteria, accelerating the death rate of the bacteria [19]. The potent inhibitory effect on tuberculosis shows the potential of these particles as a functional novel alternate drug in tuberculosis therapy.

Carbon dots (CDs) were recently demonstrated to have visible light-activated antimicrobial activity toward bacteria. The surface functionalities of CDs play a dominant role in governing their light-activated antimicrobial activity. Polymers of 2,2-(ethylenedioxy)bis(ethylamine) (EDA)-modified CDs (EDA CDs) were shown to be more effective than 3-ethoxypropylamine (EPA)-functionalized CDs in inhibiting the growth of *B. subtilis*, which was attributed to the positive charges from the amino groups of EDA CDs being more favorable to interactions with the bacteria [50].

## 8. Conclusions and Outlook 

QDs are some of the most promising nanomaterials due to their unique physical and chemical properties. To increase the antimicrobial activity of QDs, polymers are used to improve their attachment onto microorganisms. Functionalized QDs can insert into the cell wall/membrane to destroy the cell barrier, as well as produce a large amount of ROS, thereby damaging the cell components. The basic antimicrobial principle of QDs involves their electron charge transfer property and the easy availability of a π electron conjugation system with the biomatrices of microbes. Moreover, QDs are exceptional oxidizing agents, which produce free radicals in microbial organisms.

Compared with functionalized nanoparticles, QDs with quantum size are generally more effective against microorganisms than metal nanoparticles (ranging in size from 30 nm to 100 nm) [51]. For example, Kailasa et al. reported that silver nanoparticles inhibited microbial growth with MICs ranging from 18 μg/mL to 512 μg/mL [52], whereas silver QDs 7 nm in size exhibited more than 95% microbial growth inhibition with an MIC of 15 μg/mL in 3 h [48]. Moreover, most QDs are human-friendly, and they do not cause cell toxicity to human cells. However, some nanoparticles, including CdSe, TiO_2_, Fe_2_O_3_, and silver nanoparticles, were found to induce cell toxicity in human umbilical veins, endothelial cells, and lung epithelial cells [53]. Therefore, QDs are more promising than nanoparticles in antimicrobial application.

However, while most studies emphasized the antibacterial effect of QDs, few tested their antifungal activity. Thus, a broad set of antifungal studies should be conducted using QDs to facilitate the treatment of fungal diseases. In addition, further research on polymer-modified QDs can help design drugs with broad-spectrum antimicrobial activity and low cytotoxicity for animals. In summary, functionalized QDs have broad application in the antimicrobial field.

## Figures and Tables

**Figure 1 polymers-11-01670-f001:**
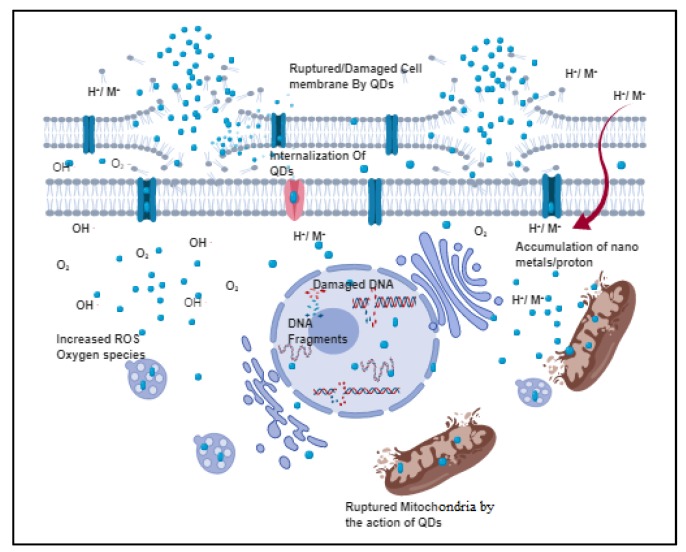
General antimicrobial mechanism of quantum dots (QDs). QDs produce antimicrobial effects through destroying cell walls/membranes, inducing free radicals, binding with genetic material, and inhibiting energy production.

**Figure 2 polymers-11-01670-f002:**
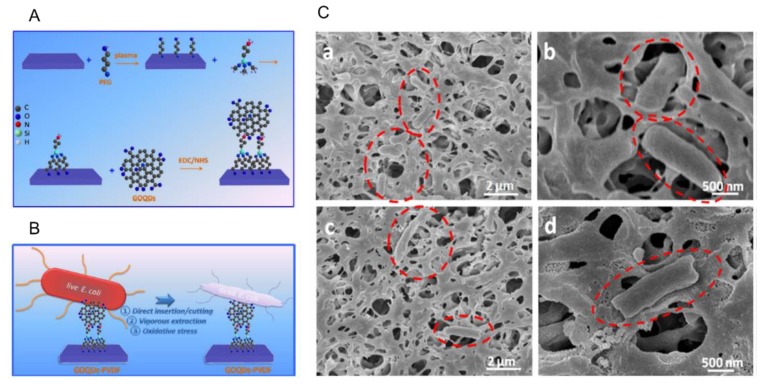
Antibacterial activity of polyvinylidene fluoride (PVDF)-modified graphene oxide QDs (GOQDs–PVDF). (**A**) Schematic illustration for covalent immobilization of GOQDs onto the PVDF membrane surface. (**B**) Bactericidal mechanism of GOQDs–PVDF. (**C**) SEM images for *Escherichia coli* at the surface of the PVDF membrane (**a**,**b**) and GOQDs–PVDF membrane (**c**,**d**). The bacteria (red circle) maintain their original structure (**a**,**b**), while they become compromised on the GOQDs–PVDF membrane (**c**,**d**). The images were reproduced with permission from Reference [26], Nature 2016.

**Figure 3 polymers-11-01670-f003:**
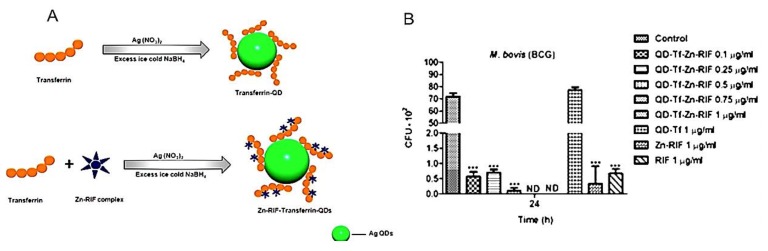
Antimicrobial effect of transferrin-modified silver QDs coupled with zinc and rifampicin (Zn/RIF/Tf QDs). (**A**) Schematic illustration of transferrin-embedded Zn/RIF complex silver QDs. (**B**) Antibacterial activity of Zn/Rif/Tf QDs. Dose-dependent killing of *Mycobacterium bovis BCG* using Zn/RIF, Tf QDs, RIF, and Zn/RIF/Tf QDs after 24 h. The bacteria were incubated with different concentrations of Zn-RIF, Tf QDs, RIF, and Zn/RIF/Tf QDs, and their survival rate was determined at the indicated time points. Media containing bacteria alone were used as control, and the corresponding concentrations of RIF and Zn(NO_3_)_2_ present in respective doses of the Zn/RIF complex were used as rifampicin and Zn(NO_3_) controls, respectively. The images were reproduced with permission from Reference [19], Nature 2016.

**Table 1 polymers-11-01670-t001:** List of quantum dots (QDs) and their mechanisms of inhibitory action on bacteria and fungi.

Quantum Dots (QDs)	Size (nm)	Microorganism (Bacteria and Fungi)	Main Mechanism of Inhibitory Action	References
Graphene QDs	3–8 nm	*Escherichia coli, Staphylococcus aureus*	Production of free radicals to damage the cell wall	[7]
Polyethyleneglycol (PEGylated) silver graphene QDs	5–8 nm	*Pseudomonas aeruginosa, Staphylococcus aureus*	The PEG-Ag graphene QDs bind with the thiol of enzymes and proteins of the cell wall/membrane, leading to leakage of cell metabolites	[8]
CdTe QDs	5–10 nm	*Escherichia coli*	The QDs insert into the cell membrane to cause membrane stress; furthermore, the heavy metal ions are released into the cells to decline the gene expression of superoxide dismutase (SOD)	[9]
CdTe–Rocephin QD complex	3 nm	*Escherichia coli*	Rochepins damage the cell wall and make pits in the membrane; then, CdTe QDs enter the cell cytoplasm, and attach to the nucleic material, preventing the gene expression of anti-oxidase	[10]
3-mercaptopropionic acid (MPA) -capped CdTe QDs	1–10 nm	*Salmonella typhimurium, Acinetobacter baumanni, Pseudomonas aeruginosa*	The QDs attach to the phospholipid layer of bacteria; also, Cd^2+^ disrupts the cellular pathway and retards cell respiration	[11]
CdSe QDs	7 nm	*Pseudomonas aeruginosa*	Internalization of Cd^2+^ cause cell toxicity and genomic toxicity	[12]
thioglycolic acid (TGA) and mercapto-acetohydrazide (TGH) lysine-capped CdSe QDs	8 nm	*Staphylococcus aureus*	Increased toxicity of Cd^2+^ causes cell death	[13]
CdSe QDs/TiO_2_/nano graphene sheets	10 nm	*Escherichia coli*	The delocalized photogenerated π electrons create oxidative stress	[14]
CdS/Ag_2_S QDs	2–19 nm	*Escherichia coli,* *Pseudomonas aeruginosa, Staphylococcus aureus*	The QDs penetrate through the cell wall and attach to DNA; then, the DNA molecules get condensed and damage DNA structure	[15]
ZnO QDs	3–7 nm	*Escherichia coli*	The photoexcitation of ZnO QDs produces an electron–hole pair; then, the electron trapped by the oxygen induces excessive reactive oxygen species (ROS)	[16]
Polyvinyl pyrrolidone (PVP)-capped ZnO QDs	2–10 nm	*Listeria monocytogenes, Salmonella enteritis, Escherichia coli*	The QDs penetrate through the cell membrane and cause cell organelle damage	[17]
Ag/In/S QDs	9.5–10 nm	*Candida albicans*	Promotion of ROS production	[18]
Zn/rifampicin/Tf QDs	10 nm	*Mycobacterium smegmatis, Mycobacterium bovis BCG*	Cell toxicity resulting in apoptosis	[19]

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
