# Peer review of "Antimicrobial Activity and Mechanism of Functionalized Quantum Dots"

_polymers, 2019, doi:10.3390/polym11101670_

Round 1

Reviewer 1 Report

This manuscript reviews development of functionalized quantum dots with efficient antimicrobial activity.  It covers a wide range of quantum dot materials from the well-known CdTe to semiconductive polymers.  Table 1 nicely summarizes the works in the past and thus very useful in the field.  I will recommend publication of this review article after the authors address the following points:

1.      Characteristics of functionalized nanoparticles other than quantum dots such as widely used gold and polystyrene nanoparticles can be included in the review along with a discussion on their antimicrobial activities compared to those of quantum dot systems.

2.      In Table 1, it says “Microorganism (Pathogens)” at the top while there is only one item in parenthesis, “(fungi)”.  Does this mean that only pathogen among the listed microorganisms is fungi?

3.      English needs to be polished.  The first sentence in the introduction stating “the challenge for human wellbeing is raising bacterial and fungal infections,” sounds like people need to increase the chance of infection by pathogens instead of suppressing it.  In abstract, “The mainly antimicrobial mechanism …” should be ”The main antimicrobial mechanism …”.  “… highly stability, exceptional … ” on page 1 should also be “… high stability, exceptionally …”.  There are many grammatical errors throughput the manuscript that need to be corrected before publication.

4.      In Table 1, “ODs” should be “QDs”.  In Fig. 1, “Mitchrondia” should be “Mitochondria”.  On top of the figure, there is an unknown small triangle that may better be removed.

Reviewer 2 Report

Dear Authors,

The manuscript ID: polymers-605620-v1 entitled “Antimicrobial activity and mechanism of functionalized quantum dots” written by Keerthiga Rajendiran, Zizhen Zhao, Desheng Pei and Ailing Fu contains some interesting data on activity of quantum dots against bacteria and fungi.

However, I have some objections to this review. Please use further literature and expand your publication for additional informations. The manuscript would be more valuable and even more reliable thanks to them.

I have a lot of suggestions in order to improve paper. These are only some of them:

Line 26: infections.[1] – infections [1].

Line 26: The – the

Line 31: imaging [2,3]. – imaging [2,3].

Line 44: components.[6] – components [6].

Line 47: Table 1. List of QDs and its mechanism on bacteria and fungi. – Table 1. List of QDs and its mechanism of inhibitory action on bacteria and fungi.

Staphylococcus Aureus – Staphylococcus aureus

Candida Albicans – Candida albicans

Table 1. Italic, dots, no dot,

Lines 84 and 85: and – and

Lines 87 and 89: PEGlyated – PEGylated

Lines 120-121: both in in vitro and in vivo conditions – both in vitro and in vivo conditions

Line 124: with 100μg/ml – with 100 μg/mL

Line 125: in vivo – in vivo

Line 139: 4.1. Mechanism of the QDs against multi-drug resistant strains – Mechanism of the QDs against multi-drug resistant bacterial strains

Line 149: Bacillus substilis – Bacillus subtilis

Line 160: μg/ml – μg/mL

Line 182: Klebsiella, and – Klebsiella pneumoniae, and

Line 187: [40]. – [40].

Line 229: [42]. – [42].

Line 236: Antibacterial activity – Antimicrobial activity

Line 270: MCs – MICs

Lines 270-271: Microsporum gypseum, Microsporum canis, Trichophyton mentagrophyte, Candida albicans, and Candida tropicalis - please use a short name, eg. M. gypseum, …

Line 277: Candida albicans – please use a short name

Line 285: E-coli – E. coli

Line 289: [49]. – [49].

Line 290: Mycobacterium smegmatis and Mycobacterium bovis-BCG – please use a short name

Line 308: Bacillus subtilis – please use a short name

Line 304: Carbon dots – different fonts

Please correct a lot of mistakes in the References.

With highest regards,

Round 2

Reviewer 2 Report

Dear Authors,

I think that some adjustments of the manuscript ID: polymers-605620-v1 entitled “Antimicrobial activity and mechanism of functionalized quantum dots” have been made. I was counting on a lot more additional information, but I accept this text after minor revision.

Please correct small mistakes:

Line 118: in in – in

Line 122: 100μg/ml – 100 μg/ml

Lines 263 and 268: T. mentagrophyte – T. mentagrophytes

Moreover, please correct the references in accordance with the instructions for authors.

With highest regards,

Author Response

Thanks for the reviewer’s carefulness.  We revised language errors and reference formats according to the reviewer’s suggestion.